# Crystallization of Long-Spaced Precision Polyacetals III: Polymorphism and Crystallization Kinetics of Even Polyacetals Spaced by 6 to 26 Methylenes

**DOI:** 10.3390/polym13101560

**Published:** 2021-05-13

**Authors:** Stephanie F. Marxsen, Manuel Häußler, Stefan Mecking, Rufina G. Alamo

**Affiliations:** 1Department of Chemical and Biomedical Engineering, FAMU-FSU College of Engineering, 2525 Pottsdamer St, Tallahassee, FL 32310, USA; sfm13@my.fsu.edu; 2Department of Chemistry, University of Konstanz, Universitätsstraße 10, 78457 Konstanz, Germany; manuel.haeussler@uni-konstanz.de (M.H.); stefan.mecking@uni-konstanz.de (S.M.)

**Keywords:** long-spaced polyacetals, isothermal crystallization, crystallization kinetics, polymorphism

## Abstract

In this paper we extend the study of polymorphism and crystallization kinetics of aliphatic polyacetals to include shorter (PA-6) and longer (PA-26) methylene lengths in a series of even long-spaced systems. On a deep quenching to 0 °C, the longest even polyacetals, PA-18 and PA-26, develop mesomorphic-like disordered structures which, on heating, transform progressively to hexagonal, Form I, and Form II crystallites. Shorter polyacetals, such as PA-6 and PA-12 cannot bypass the formation of Form I. In these systems a mixture of this form and disordered structures develops even under fast deep quenching. A prediction from melting points that Form II will not develop in polyacetals with eight or fewer methylene groups between consecutive acetals was further corroborated with data for PA-6. The temperature coefficient of the overall crystallization rate of the two highest temperature polymorphs, Form I and Form II, was analyzed from the differential scanning calorimetry (DSC) peak crystallization times. The crystallization rate of Form II shows a deep inversion at temperatures approaching the polymorphic transition region from above. The new data on PA-26 confirm that at the minimum rate the heat of fusion is so low that crystallization becomes basically extinguished. The rate inversion and dramatic drop in the heat of fusion irrespective of crystallization time are associated with a competition in nucleation between Forms I and II. The latter is due to large differences in nucleation barriers between these two phases. As PA-6 does not develop Form II, the rate data of this polyacetal display a continuous temperature gradient. The data of the extended polyacetal series demonstrate the important role of methylene sequence length on polymorphism and crystallization kinetics.

## 1. Introduction

Polyethylene-like materials with a low content of in-chain functional groups placed at a precise equal distance along the backbone are systems of interest. On the one hand, the precise equidistant placement of co-units allows one to study the effect of the size and polarity of the functional groups on chain folding and crystalline structures enabled by co-units’ rejection from or accommodation within the crystallites. On the other hand, co-units such as acetal, ester, or carbonate groups confer advantageous hydrolytic degradability to the chain, while simultaneously the long polyethylene-like sequence imparts crystalline and melting properties close to those of classical low density (LDPE) or linear-low density polyethylenes (LLDPE) [1,2,3,4,5].

Compared with traditional polyethylenes, long-spaced polyacetals can be synthesized from sustainable biomass and have the potential to be implemented as biorenewable alternatives to commodity polymers synthesized from fossil fuels [1,2,6]. Indeed, it has been demonstrated recently that similar long-spaced precision polyesters and polycarbonates are promising candidates toward “closing the loop”, in other words, enabling a truly circular economy of polyethylene-like plastic materials [7]. Furthermore, as thermoplastic semicrystalline polymers, processing of long-spaced polyacetals is from the melt. The kinetics of crystallization or rate of solidification is thus a critical parameter for predicting and optimizing processing conditions, and for correlating structure with ultimate performance.

In two preceding works [8,9], we studied the polymorphic transformations on heating and the isothermal crystallization kinetics of a set of polyacetals with the acetal unit precisely spaced by 12, 18, 19, or 23 backbone carbons. In the first work of the series [8], we demonstrated that rapidly crystallized systems with acetal groups spaced by an even (12, 18) or odd (19, 23) number of methylenes exhibit multiple melt-recrystallization events associated with polymorphic transformations on heating. These polyacetals also exhibit differences in melting points and a trend of the X-ray patterns with increasing temperature that differs between odd and even-spaced polyacetals. On fast cooling from the melt to room temperature, odd-spaced polyacetals develop a disordered, albeit layered, mesomorphic-like structure. On heating, the mesomorphic phase rearranges first to hexagonal crystals, which upon further heating melt and recrystallize to Form I crystals. The latter undergo the same melt-recrystallization event to the more stable Form II prior to their final melting point. The two even-spaced polyacetals studied (PA-12 and PA-18) cannot bypass the formation of Form I on fast cooling to room temperature and, therefore, develop a mixture of hexagonal and Form I crystals. On heating, the hexagonal structure transforms to Form I, and Form I melts and recrystallizes into Form II following the same behavior as the odd-spaced polyacetals.

Differences in layer spacing and wide-angle X-ray diffraction (WAXD) patterns of Form II between odd and even spaced polyacetals denote that the configuration of consecutive acetals with respect to the methylene sequence affects the staggering of the acetals in the crystallites. The change in the acetal staggering causes an alternating increase and decrease in melting temperatures between odd and even polyacetals with increasing length of the methylene spacer up to ~12 methylenes. *N*-alkanes experience an analogous even-odd effect due to a different setting of the methyl end group at the surface of the extended crystallites [10,11]. In precision polyacetals, packing of the crystalline acetal layers in even or odd spacers confers the same effect as does the methyl end-group in *n*-alkanes [3,8]. As discussed in our prior work, for CH_2_ spacing > 12, the odd-even effect on melting disappears as van der Waals interactions of CH_2_ sequences in the crystal become more prevalent to the packing structure than the interactions between acetal groups [8].

Disordered, hexagonal, Form I, and Form II crystals of long-spaced polyacetals are all layered, which indicates that the major difference between Form I and Form II may reside in the staggering of acetals in the crystallites, as found for precision polyethylenes with pendant halogens [12,13]. Under fast crystallization, precision polyethylenes with Cl or Br atoms adopt an *all-trans* planar conformation (Form I). Conversely, under slow crystallization, gauche conformers develop at bonds adjacent to the carbon with the halogen (Form II). The latter is an out-of-plane herringbone structure characterized by a drastic decrease of nucleation density and large double-banded spherulites [14].

A subsequent paper described the overall crystallization kinetics and linear spherulitic growth rates of the same odd-even precision polyacetal series with emphasis on the temperature coefficient of the crystallization rate [9]. The same polymorphs observed on heating developed under isothermal crystallization conditions. For the polyacetals studied, the crystallization rates of the kinetically favored forms (mesomorphic and hexagonal) were too fast to be measured experimentally. Moreover, the crystallization rates and heats of fusion of Form I and Form II display an unusual behavior with increasing temperature. While the overall crystallization kinetics of Form I follow the usual negative temperature coefficient, the dependence of the rate of Form II with decreasing temperature is inverted when approaching from above the narrow range of temperatures where Form I and Form II coexist. The resulting general trend is a deep minimum in the crystallization kinetics at the transition between polymorphs. At the temperature where the rate minimum occurs, the amount of crystals that develop is so low that for all effects, crystallization appears extinguished. The level of crystallinity recovers with a small change in crystallization temperature when either pure Form I or pure Form II develop [9].

Analysis of the rate data also indicated that formation of Form II in PA-12 required relatively high crystallization temperatures and, at those temperatures, crystals develop very slowly [9]. Such a trend indicates that the formation of Form II becomes more restricted with increasing content of acetal in the chain and infers a critical CH_2_ spacer length for the formation of Form II in polyacetals.

Continued efforts in the synthesis of these systems have enabled expansion of the set of evenly spaced polyacetals to a methylene length longer (up to 26) and shorter (down to 6) than the sequence length previously studied. Thus, in the present work we analyze the isothermal overall crystallization rates and polymorphic behavior of the new systems comparatively with prior data. To avoid the well-known odd-even effect on crystallization rates, only evenly spaced polyacetals are studied.

## 2. Materials and Methods

*Materials.* The general repeating unit of the polyacetals studied is [-O-CH_2_-O-(CH_2_)_x_-]_n_ with *x* = 6, 12, 18, or 26. Polyacetals were prepared from the corresponding diols and dimethoxymethane according to prior works [1,15]. The acetal groups per 100 CH_2_ in the long aliphatic sequence, molecular weight characteristics, and highest crystallization and melting peak temperatures from differential scanning calorimetry (DSC) runs at 10 °C/min are listed in Table 1. Due to poor solubility, the molecular weight distribution of PA-26 is suspected to be broader than for the rest of the polyacetals.

*Instrumental techniques.* The initial crystallization and melting behaviors and all isothermal crystallizations, were carried out via DSC using a TA Q2000 instrument connected to an intracooler to maximize heat transfer and to allow subambient temperature control. The static temperature and heat of fusion were calibrated with indium. Melting was also recorded at different heating rates to test for melting-recrystallization during heating. The samples were heated to about 30 °C above the final observed melting and held for 5 min to erase the previous thermal history, cooled at 40 °C/min to either 20 °C or −50 °C, held for 5 min and then heated at the chosen heating rate. Heating rates tested were 1, 2, 5, 10, 20, 40, and 80 °C/min.

For measurements of the overall crystallization rate, the samples were first heated to a temperature about 30 °C above the final observed melting for 5 min, and they were further cooled at a rate of 40 °C/min to the *T*_c_. The samples were held at *T*_c_ for sufficient time to record the exothermic heat flow. The crystallization rate at *T*_c_ was associated with the inverse of the peak crystallization time (1/*t*_0.5_).

*X-ray characterization.* Small-angle X-ray scattering (SAXS) and WAXD patterns were collected simultaneously using a Peltier stage for temperature control and the Bruker Nanostar diffractometer with Incoatec micro-focus X-ray source. The instrument is equipped with a HiStar 2D Multiwire SAXS detector and a Fuji Photo Film image plate for WAXD detection. The impressed plate was read with a Fuji FLA-7000 scanner. The samples were either isothermally crystallized in the DSC and the X-ray patterns collected at room temperature or melted on a hot plate and rapidly placed in the Peltier stage preset at a fixed temperature. The latter was needed for PA-6 that crystallizes below room temperature and melts around room temperature (20–30 °C). The patterns of the molten samples were used to estimate the WAXD-derived degree of crystallinity (*X*_c_).

Long-periods and thicknesses of crystalline regions were estimated from background and Lorentz-corrected SAXS patterns. Corrected SAXS intensities (*I*(*q*)) were used to obtain the normalized one-dimensional correlation function of the electron density fluctuations normal to the lamellar stacks according to [16,17],
(1)$$\gamma \left(r\right)=\frac{\int_{0}^{\infty }I\left(q\right)q^{2}\cos\left(rq\right)dq}{\int_{0}^{\infty }I\left(q\right)q^{2}dq}$$
where *q* is the scattering angle, and *r* the correlation length. The long period (*L*) and crystal thickness (*l*_c_) were obtained as described in prior works [8,18].

## 3. Results and Discussion

Figure 1a,b show crystallization exotherms from the melt and subsequent melting thermograms collected at 10 °C/min for all polyacetals. Except for PA-26, which shows a double exothermic transition, the crystallization of the polyacetals is single-peaked, while the melting displays multiple transitions. The latter is consistent with melting and recrystallization events during heating. As shown in Figure 1c, the highest peak temperatures decrease linearly with increasing acetal content and the heat of crystallization decreases from 140 to 50 J/g as the length of the methylene spacer decreases in the polyacetal series (Figure 1d).

Prior works demonstrated that crystallites of polyacetals are layered, in other words, two or more polyacetal repeating units participate in the crystallite with inter-chain staggering of the acetal groups (-O–CH_2_–O-) in planes that are about normal to the chain axis [8]. The number of layers in the crystal increases with increasing acetal content in the chain [8,9]. Since the melting point and heat of fusion decrease also proportionally to the increasing content of acetal, the trends of Figure 1c,d reflect the role of the acetal group as a defect in the crystal, restricting crystallization. With increasing acetal content in the chain, the enthalpic penalty associated with accommodation of the acetal group in the crystal increases, thus lowering the heat of crystallization and melting of these systems.

Figure 2 serves to evaluate reorganization on heating through display of melting scans at increasing heating rates of samples crystallized from the melt at 40 °C/min. Polyacetals PA-18 and PA-26 undergo one major melt-recrystallization event that is not fully suppressed even at the highest rate tested. As shown in Figure 2, melting-recrystallization is only suppressed at high heating rates for the shorter spaced polyacetals; the intensity of the highest melting peak decreases with increasing heating rate, but still remains after melting at 80 °C/min in PA-18 and PA-26.

Prior works showed for PA-12 and PA-18 that melting-recrystallization events demarcate transitions to different packing assemblies [8]. Among the four different packing assemblies found, the unit cells of the two highest temperature polymorphs are unknown, and are termed Form I and Form II crystals [8]. The features found for PA-26 in Figure 2d are similar to those of PA-18; the melting-recrystallization observed at ~76 °C for PA-26 can be associated with melting of Form I and recrystallization to Form II. The latter further melts at 88 °C. PA-12 displays two melt-recrystallization events at 55 and 63 °C that are associated with melting of hexagonal crystals and recrystallization first into Form I, and subsequently melting of Form I and recrystallization into Form II crystals. The more stable Form II further melts at 68 °C [8]. Although not as sharp, the evolution of the melting peaks of PA-6 with increasing heating rate is also consistent with a melting and recrystallization event at about 15 °C.

A feature that becomes prominent in the expanded polyacetal series studied here is the decreasing temperature range between the two highest melting peaks with decreasing length of the methylene spacer. We see this feature more clearly when comparing the thermograms scanned at 1 °C/min of Figure 2. From the longest to the shortest spaced polyacetal, the temperature gap decreases from 12 degrees to 9, 5, and 0, thus inferring that Form II crystals may not develop in PA-6. If we consider that PA-6 follows analogous polymorphic transformations as PA-12 but shifted to lower temperatures, the small endotherm at ~16 °C in Figure 2a could be associated with melting of a small fraction of hexagonal crystals that further recrystallize into Form I. A transformation from hexagonal to Form I crystals occurs for PA-12 in the temperature region of the shallow endotherm of PA-12 at ~50 °C. In summary, the data of Figure 2 infer that the formation of Form II persists for polyacetals spaced by 26 methylenes, while lack of the high temperature melt-recrystallization event in PA-6 indicates that the formation of Form II is more restricted as the methylene spacer decreases and may not form in polyacetals spaced by ≤6 methylenes.

X-ray patterns collected on heating polyacetals that were first rapidly quenched from the melt to 0 °C or lower temperatures document the transformation on heating to different polymorphs. Figure 3 shows the effect of a shorter (PA-6) or longer (PA-26) spacer on the evolution of X-ray patterns with increasing temperature. This figure also includes WAXD patterns of even-spaced polyacetals quenched at much lower temperatures than the 25 °C used in our earlier work [8].

The characteristic X-ray pattern of mesomorphic-like disordered polyacetal crystals is a broad pattern devoid of crystallographic reflections, as shown previously [8]. On heating, or with increasing crystallization temperature, the broad reflection sharpens to a pattern consistent with the hexagonal phase at *q* = 1.5 Å^−1^. On further heating the pattern transforms to one with three-four distinctive reflections (*q* = 1.35, 1.55, 1.62 Å^−1^) denoting transformation to Forms I and II. In the prior work, quenched at 25 °C, PA-12 and PA-18 displayed X-ray patterns consistent with mixed Form I and hexagonal crystals [8]. However, as shown in Figure 3, the present data indicate that on a deeper quenching to 0 °C, the longer spaced polyacetals PA-18 and PA-26 develop the disordered phase, similar to the structure developed by odd-spaced polyacetals under a milder quench to 25 °C [8].

The shorter-spaced polyacetals (PA-6 and PA-12) cannot bypass the formation of Form I even on fast quenching to 0 °C or to −10 °C as shown in panels a and b of Figure 3. In the latter, the WAXD patterns are unchanged with temperature, which is consistent with the formation of mixed hexagonal and Form I crystals on cooling at low temperatures, and formation of Form I and Form II at the two highest temperatures shown. Conversely, the disordered patterns of PA-18 and PA-26 slowly evolve into hexagonal crystals at temperatures of 50 and 70 °C, respectively (sharp reflection at 1.5 Å^−1^ in panels c and d). At higher temperatures, the patterns of PA-18 and PA-26 are consistent with those found earlier for Forms I and II, which for even-spaced polyacetals are indistinguishable, as discussed previously [8]. Only the pattern for Form I taken at 75 °C for PA-26 appears different from the rest of the series.

The range of crystallization temperatures (*T*_c_) for the formation of the two major, high temperature polymorphs (Forms I and II), and differences in melting points, is demarcated by their melting behavior after isothermal crystallization, as shown in the thermograms of Figure 4a–d. With increasing *T*_c_, a sharp increase in melting temperature is found for PA-12 (*T*_c_ = 64 °C), PA-18 (*T*_c_ = 75 °C), and PA-26 (*T*_c_ = 76.5 °C) which is highlighted by the red thermograms in panels b, c, and d. The unusual characteristic of the high temperature polymorphic transformation of long-spaced aliphatic polyacetals is the extremely narrow range of crystallization temperatures for such polymorphic change. As shown, the sharp increase in melting point at the transition from Form I to Form II occurs in less than 1 °C. The increase of melting temperature with increasing *T*_c_ for PA-6 is gradual; lack of a sharp increase in melting is consistent with a lack of formation of Form II.

The step increase in melting temperature at the transition between polymorphs is highlighted in plots of the melting temperatures versus crystallization temperature for each crystal phase (Figure 5a). Furthermore, Figure 5b shows equivalent plots of the heat of fusion after complete transformation at *T*_c_. At the low crystallization temperatures, the heat of fusion of PA-12, PA-18, and PA-26 is about 100 J/g and decreases very sharply at *T*_c_ approaching melting of Form I. Such remarkable decrease in heat of fusion at the transition between Form I and Form II, observed earlier in odd and even spaced polyacetals, is also present in the data of PA-26 as shown in Figure 5b. It is also remarkable that the low value of heat of fusion at the transition between Form I and Form II is basically unchanged with increasing crystallization time.

Our previous work correlated such unusual extinguished crystallization with large differences in nucleation energy barriers between both forms, with Form II requiring at least 3 times the energy of Form I to nucleate [9]. Such a difference may cause a frustration in the nucleation event of Form II since even at the transition, Form I will be always kinetically favored. As shown in Figure 5b, it is only at the narrow transition between Forms I and II that the heat of fusion is so low (<10% of the highest value). Beyond the narrow overlapping transition, pure Form II develops with just a 0.2 °C increase in crystallization temperature and the heat of fusion recovers to similarly high values. Albeit broader and shallower, minima in the heat of fusion are also apparent at the transitions between hexagonal and Form I, inferring a similar retardation effect of the kinetically favored hexagonal phase on nucleation and growth of Form I.

The melting data for each phase were used to estimate the equilibrium melting temperatures, *T*_m_^o^, of Forms I and II according to the Hoffman Weeks extrapolation [19]. For PA-18 and PA-26 the end of melting was taken as the equilibrium melting temperature of Form I, or highest *T*_c_ where Form I no longer forms. Figure 6 shows these data plotted versus content of acetal in the chain. The variation of the equilibrium melting temperature is a linear function of the content of acetal for both forms, and as inferred from the difference in melting peaks at low heating rates between Form I and Form II in Figure 2, the difference between *T*_m_^o^ of both forms for each polyacetal decreases with increasing acetal content. Both lines merge at a content of about 12.5%, corresponding to the acetal spaced by 8 methylenes, thus supporting the prior assertion that Form II will not develop in polyacetals spaced by a short methylene sequence.

The exothermic evolution of the heat flow with time measured by DSC characterizes the overall isothermal crystallization rates. Figure 7a–d displays a set of representative thermograms with increasing *T*_c_ for each polyacetal. As shown in Figure 7d, the exotherms of the long-spaced PA-26 follow the same peak-time inversion found in PA-18 and in odd-spaced polyacetals with increasing *T*_c_ [9]. With increasing *T*_c_, the peak crystallization time of PA-26 first increases up to a temperature of 76.8 °C, as expected with decreasing undercooling. Moreover, with just a 0.2 °C increase in *T*_c_, the crystallization time decreases quite dramatically as highlighted by the red thermograms in Figure 7c,d. Such a drastic decrease in crystallization time (less than half of the time) infers a sharp inversion of the crystallization kinetics at the transition between Form I and Form II. The rate inversion occurs from 76.8 to 77 °C in PA-26 and from 75.2 to 75.4 °C in PA-18. Thus, not only is the level of crystallinity extremely low as shown earlier, but the rate of nucleation and growth at the transition from Form I to Form II is deeply depressed, as shown in Figure 7c,d for PA-18 and PA-26.

The experimental observation of the inversion of the crystallization rate at the transition from Form I to Form II for PA-12 is more difficult. The high *T*_c_ and long crystallization times required for the formation of Form II in PA-12 preclude direct observation of the exothermic peak by DSC, and DSC kinetic data rely on the evolution of the heat of melting along the crystallization. Hence, the number of available experimental rate data for Form II of PA-12 is very limited. Furthermore, Form II does not develop at all in PA-6 as indicated earlier. The red thermograms of panels a and b of Figure 7 correspond to the transition from hexagonal to Form I. As this transition occurs in a much wider *T*_c_ interval, the effect on decreasing crystallization rate is not apparent from the evolution of exotherms with increasing *T*_c_ of Figure 7.

Figure 8 shows the temperature gradient of the overall crystallization rate for the series of even polyacetals studied. Here, the inverse of the peak crystallization time (1/*t*_0.5_) is taken as a measure of the rate constant at a fixed *T*_c_. Keeping the notation used in the earlier work, closed and open circles correspond to rates of Form I and Form II respectively [9]. Open diamonds are used for the hexagonal crystals. For all polyacetals, including much shorter PA-6 and longer-spaced PA-26, the overall crystallization rate of Form I follows the usual negative temperature coefficient with very steep slopes near the transition to Form II. Conversely, for crystallization temperatures where Form II develops in PA-18 and PA-26, the rate of Form II first increases with decreasing *T*_c_ as expected, and decreases sharply at temperatures approaching the transition to Form I, in the narrow range of temperatures where both forms overlap.

In addition to the rate inversion in transitioning from Form I to Form II at high *T*_c_, the rate data for PA-12 display a discontinuity, rather than a deep minimum, with increasing temperature at about 53 °C, or at the transition between hexagonal and Form I. No discontinuity is apparent in the rate data of PA-6 of Figure 8; an expected behavior, since such a transition likely occurs over a broad *T*_c_ range in this polyacetal. It is also conceivable that only Form I develops in the range of isothermal crystallization temperatures accessible by DSC for PA-6. The latter could not be determined by the X-ray patterns, which as shown in Figure 3a, are unchanged in the whole range of *T*_c_.

In a previous work, we associated the deep inversion found in the crystallization rate of Form II of polyacetals at temperatures approaching the transition to Form I from above with large differences in nucleation barriers between the two forms [9]. Besides differences in nucleation, a possible blocking effect to the nucleation and growth of Form II by Form I cannot be ruled out. Such blocking would explain that the extremely low heat of fusion observed at the transition temperature is independent of crystallization time. A temporal blocking by deposition of Form I on the developing nucleus or during the growth of Form II would frustrate crystallization of this form. The most probable temperature range for such blocking of nucleation of Form II would be near or in the transition temperature range, as observed in Figure 8. The latter is equivalent to the self-poisoning effect found in the crystallization rates of long-chain *n*-alkanes [20,21,22,23,24], methyl-terminated low molecular weight polyethylene oxides [25], and more recently in the linear growth rates of a series of high molar mass polyethylenes with precision bromine pendant atoms [14] and in amphiphilic molecular brushes of polyethylene oxide and *n*-alkyl side chains [26].

Poisoning has also been inferred in other systems which display minima in the temperature gradient of crystallization rates at the intersection between two polymorphic structures with different thermal stabilities [27,28]. Moreover, minima or discontinuities found in other semicrystalline polymers at temperatures within a transition between two polymorphs are often explained as a competition between primary nucleation and radial growth, or as a change in growth regime following the secondary nucleation theory [29,30,31,32,33,34,35]. When the isothermal crystallization range is extended to higher undercooling using fast scanning calorimetry (FSC), the crystallization rates often display two maxima with increasing crystallization temperature. The two maxima overlap at the intersection between two crystalline forms resulting in a minimum of the variation of the rate with temperature. Examples of the latter are studies in polyamides and isotactic polypropylene [36,37,38]. Coupled with morphological studies, the maxima were interpreted as a combined change in crystal structure and nucleation mechanisms.

The data of Figure 8 and all other evidence strengthen the fact that when a chain can assemble in two or more crystalline forms with drastic differences in chain conformation and nucleation barriers, approaching the transition between both forms, large and easily detectable effects can be observed during nucleation and growth. Such effects serve to better understand crystallization theories [14].

Due to very high nucleation density, it was not possible to measure linear growth rates of PA-26 and PA-6. However, recalling that in the prior work we did not observe a minimum in the growth rate of Form II at temperatures approaching melting of Form I from above, or at the transition between both forms [9], we concluded that the strongest effect leading to the observed minimum in the overall crystallization rate of polyacetals was the large difference in nucleation barriers between the two highest temperature crystal forms. The conclusion reached in our earlier work can be probed with the new rate data. Analysis of the kinetic data of Figure 8 for PA-18 and PA-26 following classical nucleation theory leads to an estimate of the difference in free energy barrier for nucleation between Form I and Form II. The data for Form II of PA-12 are insufficient, and have larger experimental uncertainties to carry out this analysis, and as shown earlier, Form II does not develop in PA-6. Although nucleation rate data are not available for these polyacetals, we take the half crystallization time data as representative of nucleation rate for this analysis. In the linear form, the rate of nucleation is [39,40,41,42,43],
(2)$$ln\;N=lnN_{o}-\frac{\Delta E_{D}}{RT}-\frac{\Delta F^{*}}{RT}$$

Here, $\Delta E_{D}$ is the activation energy for segmental transport. The term $\frac{\Delta E_{D}}{RT}$ is approximated by $\frac{U^{*}}{R\left(T-T_{\infty }\right)}$ [44,45,46], with *U** = 1500 cal/mol and *T*_∞_ = *T*_g_ − 30. The value of *T*_g_ is −80 °C. $\Delta F^{*}$ is the free energy barrier that must be surmounted to form a stable nucleus. For a 3-dimensional rectangular nucleus, $\Delta F^{*}=\frac{K_{g}T_{m}^{o2}}{\Delta T^{2}}$ with $K_{g}=\frac{32\sigma_{u}^{2}\sigma_{e}}{\Delta H_{u}^{2}}$ . *σ*_e_ and *σ*_u_ are the basal and lateral surface free energies, respectively, and Δ*H*_u_ is the latent heat of fusion. Inserting the expressions of $\frac{\Delta E_{D}}{RT}$ and $\Delta F^{*}$, and identifying *N* with 1/*t*_0.5_, Equation (2) becomes,
(3)$$ln\;t_{0.5}+\frac{U^{*}}{R\left(T-T_{\infty }\right)}=lnt_{0.5}{}_{o}-\frac{K_{g}T_{m}^{o2}}{RT\Delta T^{2}}$$

From this equation, the energy barrier for nucleation ($\Delta F^{*}$) can be determined from the slope ($K_{g}/R$) of the experimental $ln\;t_{0.5}$ data plotted vs. $\frac{T_{m}^{o2}}{T\Delta T^{2}}$. These plots are shown in Figure 9a,b for data of Form I and Form II of PA-18 and PA-26. The equilibrium melting temperatures for each crystalline form are those from Figure 6. Rata data at the minimum that are affected by the competition between both forms are not included in this analysis.

The negative temperature coefficient for both forms is linear, as shown in Figure 9, with slopes for Form II that are about twice the value of Form I, thus corroborating that the nucleation barrier of Form II is significantly higher. Taking for example the slopes of PA-26, for undercooling Δ*T* = 15.5 °C, the values of $\Delta F^{*}$ for Form I and Form II are 3.45 kJ/mol and 6.42 kJ/mol respectively. We should now consider that these energy barriers are estimates for primary nucleation. The expectation is that the barriers for primary nucleation will higher, or costlier, than for nucleation on the surface of an already formed nucleus or crystal. Secondary nucleation energy barriers calculated from linear growth rates are available from our prior work [9]. From those data, at the same undercooling of 15.5 degrees, we find values of $\Delta F^{*}$ for secondary nucleation of ~1 kJ/mol for Form I and ~5 kJ/mol for Form II. The latter are significantly lower than the barriers we obtained from Figure 9 for primary nucleation, as expected. More important is the fact that the computed differences either for primary or secondary nucleation result in a higher energy barrier for Form II than for Form I.

To obtain information about the evolution of level of crystallinity and crystal thickness with increasing crystallization temperature for the series of polyacetals studied, data from temperature dependent SAXS experiments are next analyzed. The evolution of Lorentz-corrected SAXS patterns as a function of increasing temperature is given in Figure 10a–d. For all polyacetals, the SAXS scattering peak shifts to lower *q* with increasing temperature indicating an increase in the correlated long spacing. Of interest is the variation of the shape of the scattering peak with increasing temperature between the longest spaced polyacetals, PA-18 and PA-26, and the shorter ones, PA-6 and PA-12.

We recall from Figure 3 that quenched to 0 °C, PA-18 and PA-26 develop mesomorphic structures, which as shown by the broad SAXS patterns of Figure 10c,d, are obviously poorly correlated. The development of hexagonal crystals in these systems at 50 and 70 °C respectively is demarcated by a drastic increase in SAXS invariant and a large shift of *q*. The scattering becomes sharper and more intense at higher crystallization temperatures when Form I and Form II develop. The crystalline acetal layer peak is prominent in the SAXS patterns of PA-26 at *q* ~ 0.2 Å^−1^ (30.5 Å). Comparing the layer peak with the *all-trans* length of the PA-26 repeating unit of 36.4 Å, gives a chain tilt with respect to the normal layer of 33°. The chain tilts found in *n*-alkanes and precision polyethylenes with halogens are basically identical to this value [12,13,47].

Given the fact that for PA-6 and PA-12 a mixture of Form I and hexagonal crystals develops even on fast quenching at the lowest *T*_c_, the SAXS patterns at the low temperatures are sharper than for the longer spaced polyacetals, and except for the shift in *q*, they change little with increasing *T*_c_. Only when Form II develops at the highest *T*_c_ is the intensity of the scattering peak higher, as shown in Figure 10b.

The structural data for levels of crystallinity (*X*_c_) extracted from the WAXD patterns after subtraction of the amorphous halo, and long periods (*L* = 2π/*q*) and lamellae thicknesses (*l*_c_) from the SAXS patterns, are given in Figure 11 as a function of crystallization temperature. The lamellae thickness was obtained by analysis of the one-dimensional correlation function applied to corrected SAXS intensities as indicated in the experimental section. The level of crystallinity is low (~36%) for PA-6 at any *T*_c_, but clearly increases with temperature for the rest of the polyacetals from ~45% to ~70% with the major increase happening in the *T*_c_ range of formation of Form I and Form II. We omitted values of *X*_c_ for mesomorphic structures of PA-18 and PA-26 due to large uncertainties in calculating *X*_c_ from broad WAXD patterns.

.

Coupled with the low levels of crystallinity, PA-6 displays larger values for the long period than those shown in Figure 11b for the other three polyacetals. However, the trend with temperature is analogous for the three longest polyacetals. Up to about 65 °C, the long period increases slowly from 80 Å to ~100 Å, and increases sharply at the transition from hexagonal to Form I, thus following the same sudden increase found at the same polymorphic transition on slow heating [8].

The variation of the lamellae thickness with temperature is given in Figure 11c, and as a function of undercooling in Figure 11d. As shown, not only the level of crystallinity of PA-6 is very low but also the lamellae thicknesses which are just 40–50 Å in the whole range of temperatures analyzed. The crystal thickness for the other three polyacetals follows the trend of the long period. The lamellae thickness increases slowly, from about 40 to 60 Å in the *T*_c_ region of mesomorphic and hexagonal crystals, and increases sharply at the transition from hexagonal to Form I. The lamellae thickness of Form I and Form II do not differ as greatly, as shown. A table collecting these data can be found in the Appendix A of this manuscript (Appendix A). As shown in Figure 11d, plotted versus undercooling, all lamellae thicknesses collapse into two lines with distinctive slopes. Low thicknesses with small temperature variation for Δ*T* > 15°, and much greater values in a small range of undercooling (Δ*T* = 10 ± 2 °C) leading to a very steep change with temperature. Only two data for PA-18 at the highest undercooling diverge from the rest.

Prior SAXS work on slow heating demonstrated that the evolution from disordered to hexagonal crystals involves a small crystal thickening, whereas the transformation via melting-recrystallization from hexagonal to crystals of Form I involves the addition in the chain axis of one extra repeating unit to the lamellae thickness [8]. The earlier findings are now probed with the new crystal thicknesses for PA-26 also obtained on heating. Taking into account the chain tilt of 33° obtained earlier for the crystals of PA-26, and an *all-trans* packing, the layer thickness normal to the acetal layer is 30 Å. Hence, the lamellae thickness of 55 Å at *T*_c_ = 70 °C for hexagonal PA-26 shown in Figure 11d corresponds to ~2 crystalline layers. The crystal thickness for Form I at 75 °C is 94 Å and corresponds to 3 crystalline layers. These data give further evidence of the quantized nature of the crystal thicknesses of polyacetals and of the sudden increase by one repeating unit at the hexagonal to Form I transition found earlier in shorter odd and even-spaced polyacetals on continuous slow heating [8]. Lamellae thicknesses close to integer values suggest a placement of acetal groups at the lamellae basal surface. A different configuration of the acetal with respect to the *all-trans* methylenes between odd and even polyacetals will change the mode of acetal staggering in the crystalline layer and at the lamellar surface. Such change explains the odd-even effect on melting observed previously [3,8].

## 4. Conclusions

Many similarities, but also important differences are found in the development with temperature of different polymorphic structures, and for the isothermal crystallization kinetics, when much shorter (PA-6) and longer (PA-26) spaced aliphatic polyacetals are included in the studies of a series of even polyacetals. The following are the most relevant conclusions:The longest even-spaced polyacetals (PA-18 and PA-26) develop disordered, mesomorphic-like structures under fast crystallization to 0 °C. The reorganization of disordered structures on heating follows the same behavior for all polyacetals. On heating, poorly organized disordered structures transform to layered hexagonal crystals, which upon further heating melt and recrystallize into Form I crystals. The latter further melt and recrystallize into Form II. The same polymorphic structures are obtained crystallizing directly from the melt.Shorter spaced polyacetals (PA-6 and PA-12) cannot bypass the formation of Form I even under fast quenching, thus developing mixed hexagonal and Form I crystals in the low temperature range. Under heating, hexagonal and Form I crystals undergo the same transformation as for the longer polyacetals.From the variation of the equilibrium melting temperatures with content of acetal groups in the chain, it is predicted that Form II will not develop in polyacetals spaced by ≤8 CH_2_. Indeed, we find that Form II does not develop in PA-6The overall crystallization rates of Form I display the usual negative temperature coefficient. However, with decreasing temperature, the rates of Form II first increase as expected and decrease drastically when approaching from above the melting of Form I, or in the narrow transition range from Form I to Form II. The unusual rate inversion is found in PA-12, PA-18 and PA-26 but not in PA-6, as predicted. The negative temperature gradient of the crystallization rates of PA-6 is continuous. The new rate data on PA-26 corroborate that crystallization is practically extinguished at *T*_c_ where the rate minimum occurs.Analysis of the rate data according to classical nucleation theory give at least double values for the energy barrier of nucleation of Form II than for nucleation of Form I. The difference supports the assertion that competition in nucleation between both forms, and frustration in the formation of Form II by the kinetically favored Form I, are responsible for the observed rate minima and for the low heat of fusion at *T*_c_ where the rate minima are found.The level of crystallinity and crystal thicknesses are low (40–50 Å), and change slowly in the low *T*_c_ region where disordered and hexagonal crystals are formed. Crystallinity and lamellae thicknesses increase sharply at the transition from hexagonal to Form I. As found earlier in shorter spaced polyacetals [8], upon transformation from hexagonal to Form I crystals, the core crystal thickness of PA-26 increases by one repeating unit, while the transition from Form I to Form II undergoes a small increase in crystal thickness.

The above emphasize the importance of methylene sequence length on packing at the level of the unit cell of long-spaced polyacetals. Furthermore, analyses of the crystallization rates make relevant the effect of polymorphism on decreasing drastically the crystallization rate at the transition between the two highest temperature polymorphs.

## Figures and Tables

**Figure 1 polymers-13-01560-f001:**
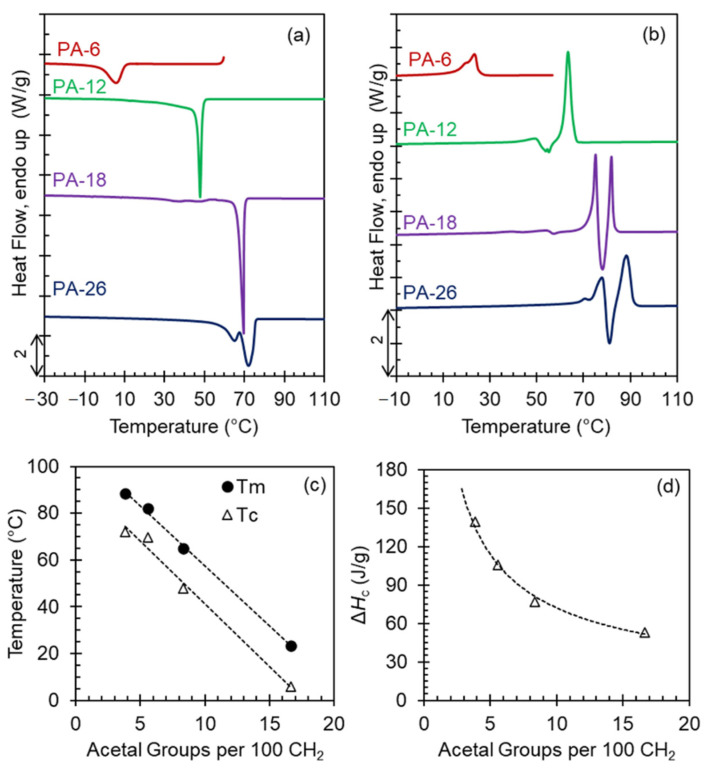
Cooling exotherms (**a**) and subsequent melting endotherms (**b**) of even polyacetals collected by differential scanning calorimetry (DSC) at 10 °C/min. Data have been vertically shifted for clarity. (**c**) Highest peak crystallization (open triangles) and melting (filled circles) temperatures, and (**d**) heat of crystallization as a function of the number of acetal groups per 100 methylene groups in the aliphatic spacer.

**Figure 2 polymers-13-01560-f002:**
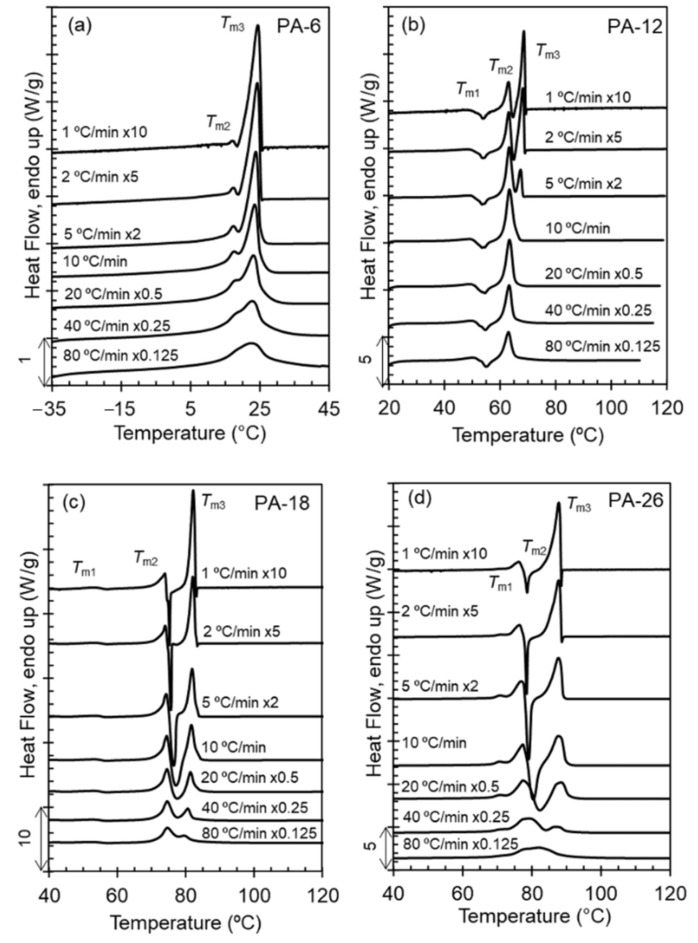
Melting endotherms collected on heating at the indicated rates for (**a**) PA-6, (**b**) PA-12, (**c**) PA-18, and (**d**) PA-26. Prior to melting, samples were cooled from the melt at 40 °C/min. Data have been normalized by heating rate and vertically shifted for clarity.

**Figure 3 polymers-13-01560-f003:**
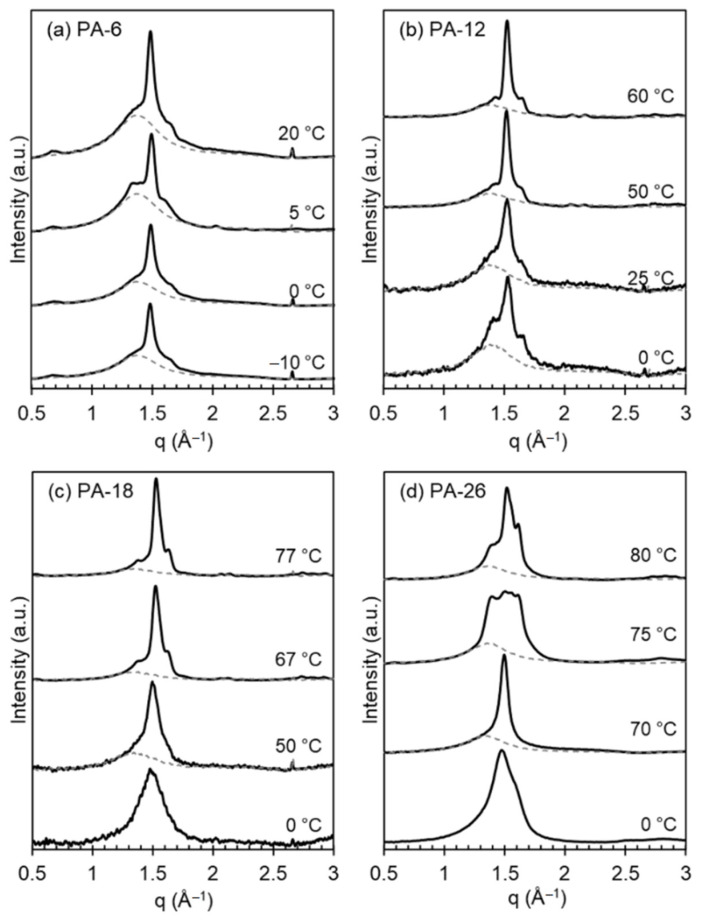
Wide-angle X-ray diffractograms (WAXD) for (**a**) PA-6, (**b**) PA-12, (**c**) PA-18, and (**d**) PA-26 crystallized or quenched at the temperatures indicated. Data have been vertically shifted for clarity. Dashed gray lines representing the experimental amorphous halo.

**Figure 4 polymers-13-01560-f004:**
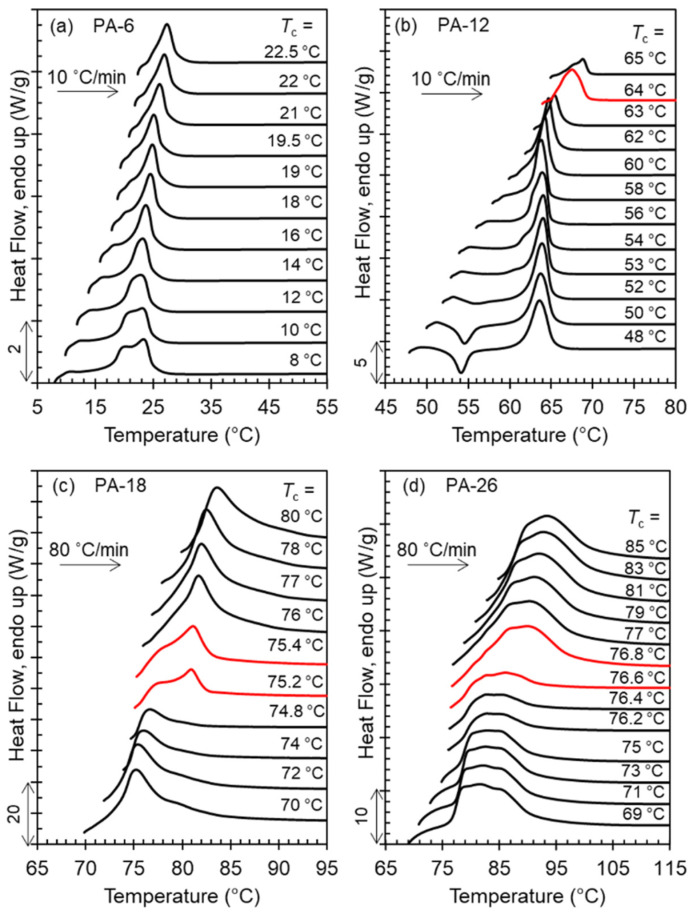
Melting endotherms after isothermal crystallization at indicated *T*_c_ for (**a**) PA-6, (**b**) PA-12, (**c**) PA-18, and (**d**) PA-26. Melting-recrystallization was minimized by heating at a rate of 80 °C/min for PA-18, and PA-26. The transition from Form I (low melting) to Form II (high melting) is indicated in PA-12, PA-18, and PA-26 by red thermograms. Data have been vertically shifted for clarity.

**Figure 5 polymers-13-01560-f005:**
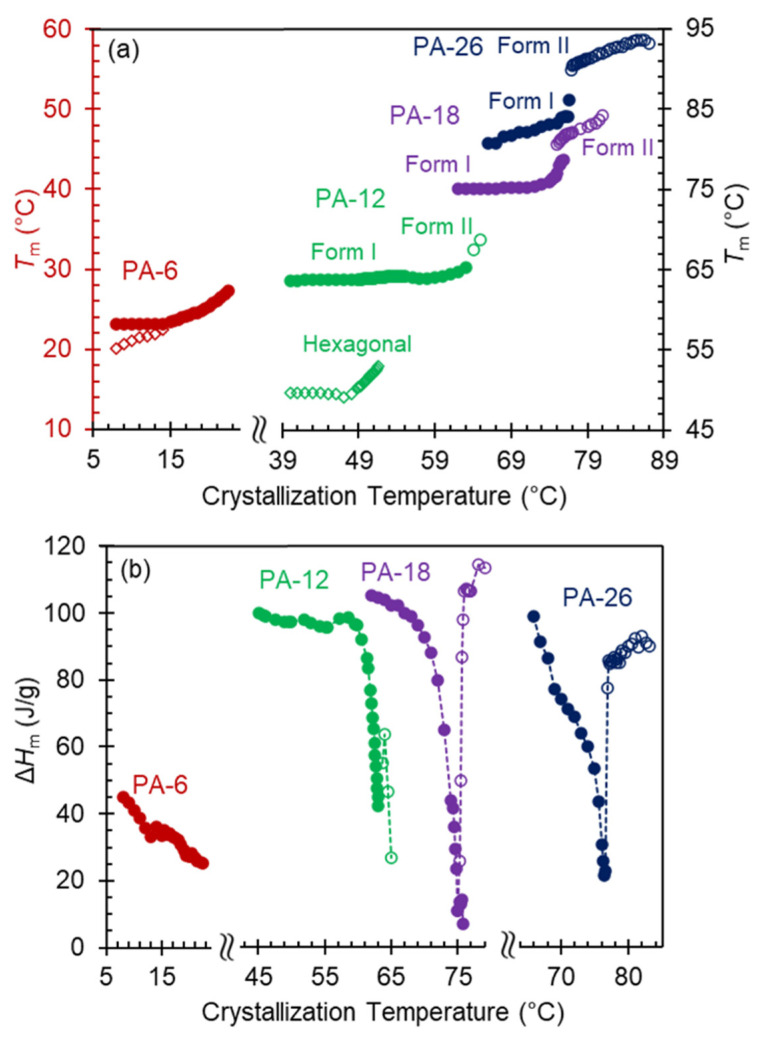
(**a**) Melting temperatures, and (**b**) heat of fusion after complete transformation as a function of crystallization temperature for PA-6 (red), PA-12 (green), PA-18 (purple), and PA-26 (dark blue). Filled circles correspond to Form II, open circles to Form I, and open diamonds to hexagonal crystals. Heat of fusion data are joined by dotted lines to emphasize the extremely low heat of fusion at the transition between Form I and Form II.

**Figure 6 polymers-13-01560-f006:**
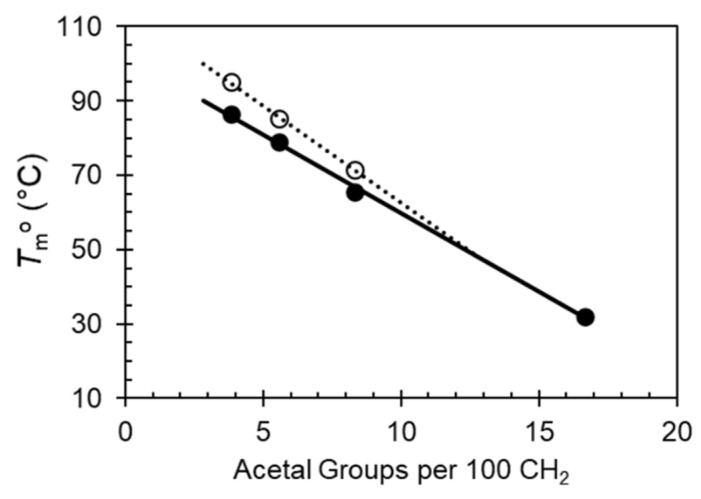
Equilibrium melting temperatures of even-spaced polyacetals plotted versus number of acetal groups per 100 methylene groups in the aliphatic spacer. Open circles are data for Form II and closed circles are data for Form I.

**Figure 7 polymers-13-01560-f007:**
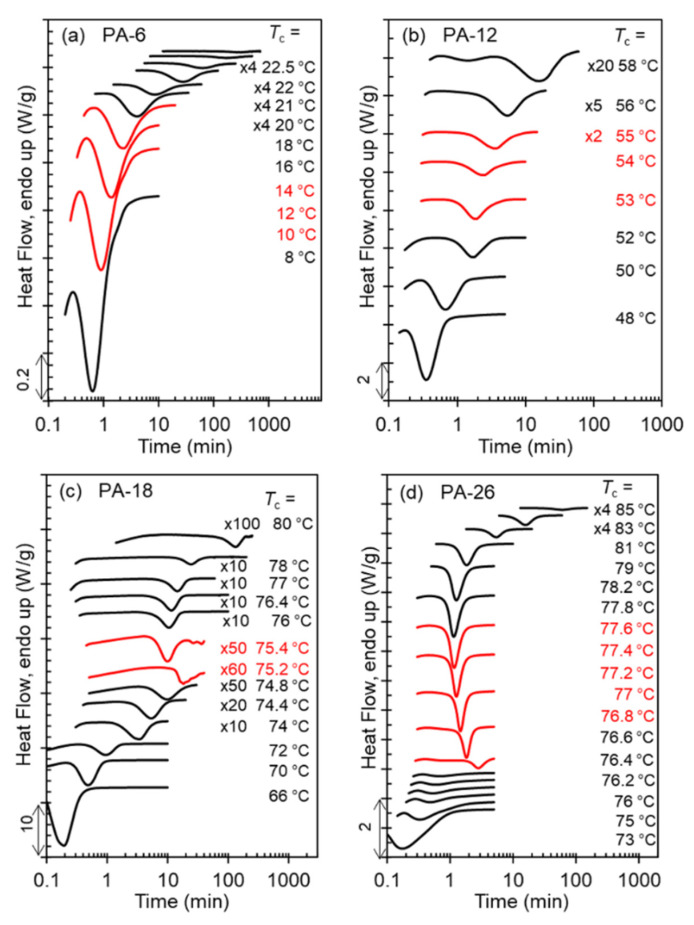
Crystallization exotherms collected as a function of time during isothermal crystallization at the indicated *T*_c_ for (**a**) PA-6, (**b**) PA-12, (**c**) PA-18, and (**d**) PA-26. Data have been vertically shifted for clarity. For PA-18 and PA-26, red thermograms indicate the transition between Form I and Form II. For PA-6 and PA-12 the red thermograms infer transition from hexagonal to Form I. Where indicated, data were multiplied by the constant factors shown to better display small heat flow.

**Figure 8 polymers-13-01560-f008:**
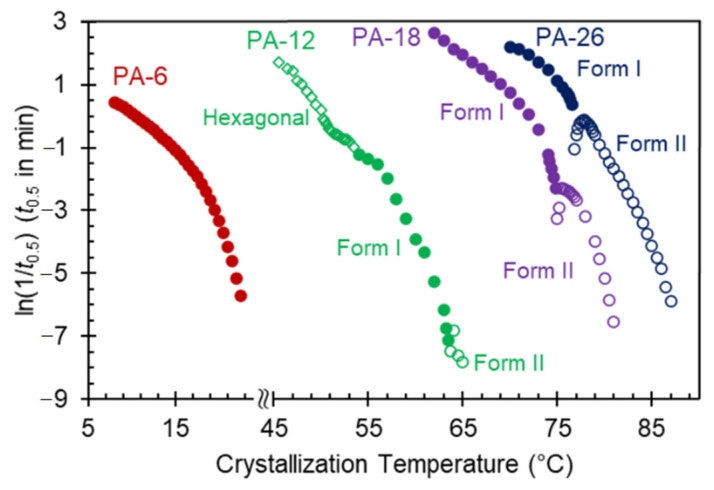
Overall rate of crystallization shown as the natural logarithm of the inverse half-crystallization time (1/*t*_0.5_) as a function of crystallization temperature for PA-6 (red), PA-12 (green), PA-18 (purple), and PA-26 (dark blue). Filled circles correspond to Form II, open circles to Form I, and open diamonds to hexagonal.

**Figure 9 polymers-13-01560-f009:**
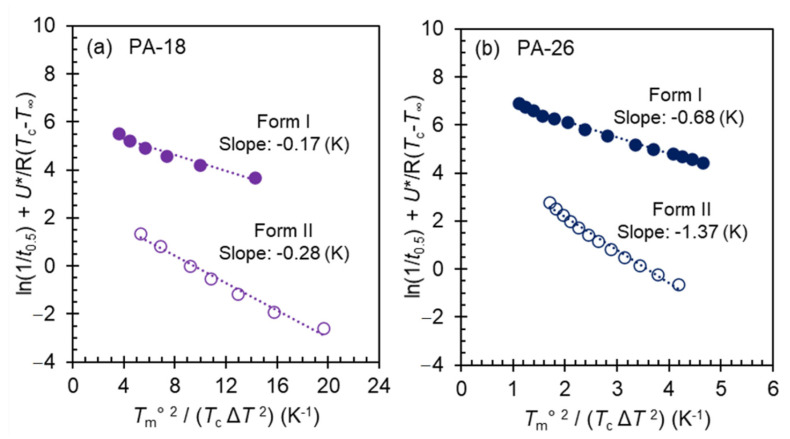
Analysis of the temperature coefficient of the crystallization rate according to classical 3D nucleation theory (Equation (3)) for long-spaced polyacetals PA-18 (**a**) and PA-26 (**b**). Closed symbols are Form I and open symbols are Form II.

**Figure 10 polymers-13-01560-f010:**
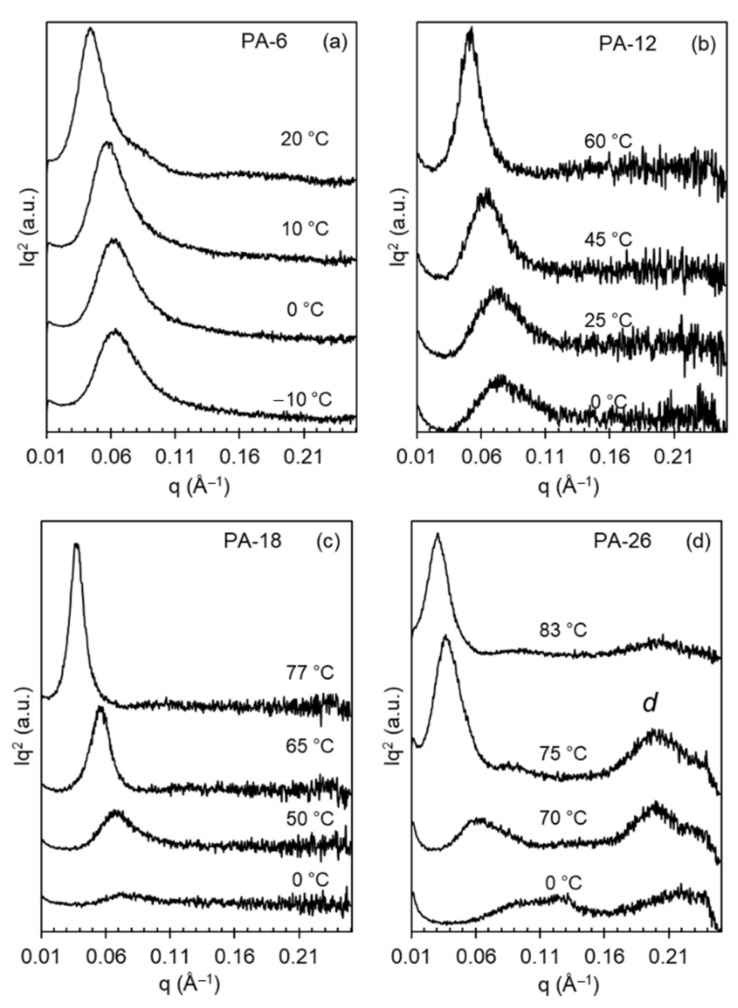
Lorentz-corrected small-angle X-ray scattering patterns for (**a**) PA-6, (**b**) PA-12, (**c**) PA-18, and (**d**) PA-26 collected on heating to the indicated temperatures. *d* denotes the crystalline acetal layer peak. Data have been vertically shifted for clarity.

**Figure 11 polymers-13-01560-f011:**
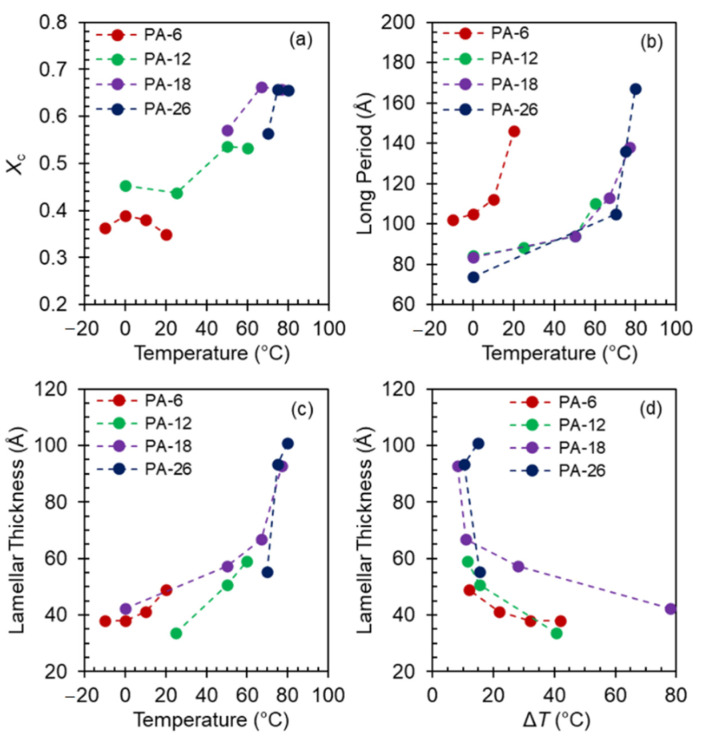
(**a**) Degree of crystallinity (*X*_c_) from WAXD patterns, (**b**) long period, and (**c**) lamellae thicknesses vs. temperature collected on heating to the indicated temperatures. (**d**) Lamellae thicknesses vs. undercooling ΔT

**Table 1 polymers-13-01560-t001:** Molecular mass characterization of even long-spaced polyacetals.

Sample	Acetal Groups Per 100 Aliphatic CH_2_	*M*_n_ (kg/mol)	*M*_w_/*M*_n_	*T*_m_ (°C)	*T*_c_ (°C)
PA-6	16.7	14.2 ^a^	2.0 ^a^	23.4	5.9
PA-12	8.33	30.2 ^b^	2.3 ^b^	63.4	47.7
PA-18	5.56	16.0 ^c^	-	82.0	69.6
PA-26	3.85	- ^d^	- ^d^	88.3	72.3

^a^ Determined by GPC at 40 °C in THF versus polystyrene standards. ^b^ Determined by GPC at 160 °C in 1,2,4-trichlorobenzene versus polyethylene standards. ^c^ Determined by end-group analysis from ^1^H nuclear magnetic resonance (NMR) spectroscopy. ^d^ Not soluble in THF, and degraded in dichlorobenzene at 160 °C.

## Data Availability

Data are contained within the article.

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
