# Peer review of "Crystallization of Long-Spaced Precision Polyacetals III: Polymorphism and Crystallization Kinetics of Even Polyacetals Spaced by 6 to 26 Methylenes"

_polymers, 2021, doi:10.3390/polym13101560_

Round 1

Reviewer 1 Report

The manuscript is quite well organized and data/results reasonably analysed and thoroughly presented. There are some minor comments:

  1. Conclusion texts may be too lengthy. They can be condensed/shortened for more focus of the novelty and contribution.
  2. Phrases in Conclusions may need refinement and condensation. Using Conclusion-1 as an example. “1-The formation of disordered, mesomorphic-like structures under fast crystallization occurs not just in odd but also in the longest even-spaced polyacetals (PA-18 and PA26). The reorganization of disordered structures on heating follows the same behavior for all polyacetals. On heating, poorly organized disordered structures transform slowly to layered hexagonal crystals, which on further heating melt and recrystallize into Form I crystals. The latter further melt and recrystallize into Form II. The same polymorphic structures are obtained crystallizing directly from the melt.” Authors’ mentioning of “..occurs not just in odd.., but in even…”  This work did not deal with “odd”, why comparing odd to even? Secondly, is there result showing “..transform slowly to layered hexagonal crystals”?. Thirdly, “..are obtained crystallizing directly from the melt.” Is complete?  All other conclusion texts need similar attention.
  3. In discussion texts, authors expressed: “Prior works demonstrated that crystallites of polyacetals are layered, in other words, two or more repeating units participate in the crystallite with inter-chain staggering of the acetal groups in planes that are about normal to the chain axis.” Two questions: (1) where is reference for the crystallites being layered?” (2) if crystallites are layered, how can it be deduced that “two or more repeating units participate in the crystallite with inter-chain staggering of the acetal groups in planes that are about normal to the chain axis”?  That is, what is relationship between the “layered crystallites” and “two repeating units participate in the crystallites”?
  4. Line 195. Statement of ”Prior works showed for PA-12 and PA-18 that melting-recrystallization events demarcate transitions to different packing assemblies.” Where are “prior works”?
  5. The article title of “Long-Spaced Precision Polyacetals” does not appear to be clear in meaning? Did’nt author say that they studied long-spaced as well as “short space”? So, why only “long-spaced” in title? “long-spaced” by itself does not convey the meaning of structure clearly. In addition, what is the meaning of “precision” in title?

Author Response

First review

  1. We thank the reviewer for his/her comments on a manuscript well organized.
  2. The reason we wrote the conclusions as relative short paragraphs is precisely to emphasize the new aspects revealed by this work. Nonetheless, we have significantly condensed two of the initial paragraphs and removed the specific reference of add polyacetals. We kept the rest because we could not find a good way to condense them further while keeping a comprehensive conclusion. Specifically, we have condensed the first paragraph from:
  3. Initial manuscript: The formation of disordered, mesomorphic-like structures under fast crystallization occurs not just in odd but also in the longest even-spaced polyacetals (PA-18 and PA-26). The reorganization of disordered structures on heating follows the same behavior for all polyacetals. On heating, poorly organized disordered structures transform slowly to layered hexagonal crystals, which on further heating melt and recrystallize into Form I crystals. The latter further melt and recrystallize into Form II. The same polymorphic structures are obtained crystallizing directly from the melt.

To

  • Revised manuscript: The longest even-spaced polyacetals (PA-18 and PA-26) develop disordered, mesomorphic-like structures under fast crystallization to 0°C. The reorganization of disordered structures on heating follows the same behavior for all polyacetals. On heating, poorly organized disordered structures transform to layered hexagonal crystals, which on further heating melt and recrystallize into Form I crystals. The latter further melt and recrystallize into Form II. The same polymorphic structures are obtained crystallizing directly from the melt.

And in second paragraph we remove a superfluous paragraph that was in the template and shortened this conclusion substantially:

  1. Initial manuscript: Shorter spaced polyacetals (PA-6 and PA-12) crystallize at lower temperatures and cannot bypass the formation of Form I even under fast quenching, thus developing mixed hexagonal and Form I crystals in the low temperature range. Under heating, hexagonal and Form I crystals undergo the same transformation as for the longer polyacetals. Authors should discuss the results and how they can be interpreted from the perspective of previous studies and of the working hypothesis.

To

  • Revised manuscript: Shorter spaced polyacetals (PA-6 and PA-12) cannot bypass the formation of Form I even under fast quenching, thus developing mixed hexagonal and Form I crystals in the low temperature range. Under heating, hexagonal and Form I crystals undergo the same transformation as for the longer polyacetals.

 We have also removed the word “slowly” (first paragraph) and made a small change on the 4th paragraph

(1) As requested, we have added the appropriate reference on line number 196, and (2) reworded the paragraph on lines 175-178 to clarify that the repeating unit comprises both the acetal group (-O-CH2-O-) and the methylene run (-CH2)x- The repeating unit was defined on line 118 of the experimental section. This paragraph now reads: “Prior works demonstrated that crystallites of polyacetals are layered, in other words, two or more polyacetal repeating units participate in the crystallite with inter-chain staggering of the acetal groups (-O-CH2-O-) in planes that are about normal to the chain axis [8].”

  1. The appropriate reference is now added to the paragraph of line 195.
  2. We believe the title is correct. This is the third part of a series of publications describing details of the crystallization of long-spaced polyacetals. The reason the word “precision” is added to the title is to emphasize that the long CH2 run between consecutive acetal groups is of a precise length. As mentioned above, the repeating unit is clearly defined in the experimental part, in line 118. All polyacetals we studied are long-spaced. In analyzing and discussing the data, we refer to the shortest CH2 run length and the longest CH2 run length studied.

Reviewer 2 Report

This is a well written paper that follows a series of works by Alamo et al. in even spaced polyacetals. The paper can be accepted as is or with minor revision.

1) What is the effect  of Mn? In Table 1, PA-12 has twice the molecular weight as PA-6 and PA-18, while the Mn value of PA-26 is unknown.

2) In Figure 1, there seems to be some minor exothermic events (Fig 1a) in between 30-55 ºC for PA-18, which seem to correspond with a low temperature endothermic signal in a similar temperature range (Fig 1b). Are these relevant or just noise?

3) In Figure 6, if we extrapolate the equilibrium melting temperature to 0 acetal groups we should obtained the value of PE, but the data seems to extrapolate to much lower values, why?

Author Response

We also thank the reviewer for the nice comments on our manuscript

  1. Some effect of Mn on melting for any given polyacetal is expected. It is also expected that the effect will saturate at some molar mass, which is unknown for these polymers. From the data of Figure 6, one could say that in the range of Mn studied, 15K < Mn < 30K, the effect of Mn on melting is most probably not very large.
  2. Yes, there are minor exotherms in the range of 30-55°C for PA-18. They could be some low molar mass components crystallizing at a lower temperature. We believe they are not affecting any of the data or conclusions reached in this manuscript.
  3. One should notice that the unit cells of Form I and Form II are expected to be quite different than the orthorhombic unit cell of polyethylene. A linear extrapolation to the Tmo of polyethylene, 145.5 °C is not expected. Notice that our extrapolation is for melting data of polyacetals crystals with the same crystal structure (same WAXD pattern, either Form I or Form II)